# PON1 Status in Relation to Gulf War Illness: Evidence of Gene–Exposure Interactions from a Multisite Case–Control Study of 1990–1991 Gulf War Veterans

**DOI:** 10.3390/ijerph21080964

**Published:** 2024-07-24

**Authors:** Lea Steele, Clement E. Furlong, Rebecca J. Richter, Judit Marsillach, Patricia A. Janulewicz, Maxine H. Krengel, Nancy G. Klimas, Kimberly Sullivan, Linda L. Chao

**Affiliations:** 1Veterans Health Research Program, Yudofsky Division of Neuropsychiatry, Department of Psychiatry and Behavioral Sciences, Baylor College of Medicine, Houston, TX 77030, USA; lea.steele@bcm.edu; 2Department of Medicine (Division Medical Genetics), University of Washington, Seattle, WA 98195, USA; clem@uw.edu (C.E.F.); rrichter@uw.edu (R.J.R.); 3Department of Genome Sciences, University of Washington, Seattle, WA 98105, USA; 4Department of Environmental and Occupational Health Sciences, University of Washington, Seattle, WA 98105, USA; jmarsi@uw.edu; 5Department of Environmental Health, Boston University School of Public Health, Boston, MA 02118, USA; paj@bu.edu (P.A.J.); tty@bu.edu (K.S.); 6Department of Neurology, Boston University School of Medicine, Boston, MA 02118, USA; mhk@bu.edu; 7Dr. Kiran C. Patel College of Osteopathic Medicine, Institute for Neuroimmune Medicine, Nova Southeastern University, Fort Lauderdale, FL 22238, USA; nklimas@nova.edu; 8Geriatric Research Education and Clinical Center, Miami Veterans Affaris Medical Center, Miami, FL 22125, USA; 9Department of Radiology and Biomedical Imaging, University of California, San Francisco, CA 94142, USA; 10Department of Psychiatry and Behavioral Sciences, University of California, San Francisco, CA 94142, USA; 11San Francisco Veterans Affairs Health Care System, 4150 Clement Street (114M), San Francisco, CA 94121, USA

**Keywords:** PON1, Gulf War illness, Gulf War veterans, deployment-related exposures, organophosphate, pesticides, pyridostigmine bromide, gene–environment interaction

## Abstract

Background: Deployment-related neurotoxicant exposures are implicated in the etiology of Gulf War illness (GWI), the multisymptom condition associated with military service in the 1990–1991 Gulf War (GW). A Q/R polymorphism at position 192 of the paraoxonase (PON)-1 enzyme produce PON1_192_ variants with different capacities for neutralizing specific chemicals, including certain acetylcholinesterase inhibitors. Methods: We evaluated PON1_192_ status and GW exposures in 295 GWI cases and 103 GW veteran controls. Multivariable logistic regression determined independent associations of GWI with GW exposures overall and in PON1_192_ subgroups. Exact logistic regression explored effects of exposure combinations in PON1_192_ subgroups. Results: Hearing chemical alarms (proxy for possible nerve agent exposure) was associated with GWI only among RR status veterans (OR = 8.60, *p* = 0.014). Deployment-related skin pesticide use was associated with GWI only among QQ (OR = 3.30, *p* = 0.010) and QR (OR = 4.22, *p* < 0.001) status veterans. Exploratory assessments indicated that chemical alarms were associated with GWI in the subgroup of RR status veterans who took pyridostigmine bromide (PB) (exact OR = 19.02, *p* = 0.009) but not RR veterans who did not take PB (exact OR = 0.97, *p* = 1.00). Similarly, skin pesticide use was associated with GWI among QQ status veterans who took PB (exact OR = 6.34, *p* = 0.001) but not QQ veterans who did not take PB (exact OR = 0.59, *p* = 0.782). Conclusion: Study results suggest a complex pattern of PON1_192_ exposures and exposure–exposure interactions in the development of GWI.

## 1. Introduction

Gulf War illness (GWI) is a chronic, multisymptom, multi-system disorder estimated to affect approximately 30% of the nearly 700,000 veterans who served in the 1990–1991 Gulf War (GW) [1,2,3]. Extensive research has investigated the etiology and biological nature of GWI in the decades since the GW ended. Gulf War veterans encountered a broad range of potentially hazardous exposures during deployment, as detailed in numerous studies and government reports [1,4,5,6,7,8,9]. However, population and clinical studies have consistently identified only a limited number of deployment-related exposures as the most significant risk factors for GWI [1,9]. These include the prolonged use of pesticides and insect repellants [4,8], exposure to nerve agents [10,11,12,13], and the use of pyridostigmine bromide (PB), a carbamate drug used as prophylaxis to protect troops from potential Iraqi nerve agent attacks [14,15]. Many of these prominent GWI risk factors exert toxic effects by acting acutely as acetylcholinesterase inhibitors (AChEis), reducing the breakdown of the neurotransmitter acetylcholine in the nervous system [16,17].

A longstanding question concerns why some GW veterans developed GWI after the war while others with similar deployment-related exposures did not. One possibility is that vulnerability to certain GW-related exposures differed as a result of individual variability in the biological processes that confer protection from adverse effects [18]. The Research Advisory Committee on Gulf War Veterans’ Illnesses [2] and the Institute of Medicine (IOM) [19], now the National Academy of Medicine (NAM), have called for investigation of potential genetic variability in GW veterans’ responses to deployment-related exposures.

Paraoxonase (PON)-1 is a human enzyme capable of hydrolyzing the active metabolites (oxons) of a number of organophosphorus (OP) and other xenobiotic substrates [20]. Polymorphisms in the PON1_192_ coding region (Q192R) can influence the catalytic efficiency of PON1 and affect its ability to hydrolyze different OP compounds [21,22,23]. For example, in vitro studies suggest that the PON1_192_ Q alloform is more effective than the PON1_192_ R alloform in hydrolyzing sarin (a nerve agent) [24], while in vivo knock-out mice studies clearly indicate that the PON1_192_ R alloform is more effective than the PON1_192_ Q alloform in protecting against chlorpyrifos oxon exposures (toxic oxon forms of OP compounds) [25].

A limited number of studies have evaluated PON1_192_ genotypes and alloform in relation to GWI [26,27,28]. Haley and colleagues [26,27] reported that veterans with the PON1_192_ R allele were at increased risk of having GWI/Haley syndromes, particularly if they were in locations associated with nerve agent exposures or heard chemical alarms during the war [26,27]. However, these studies did not consider the effects of other GW exposures and PON1 variants.

The current study sought to evaluate GWI risk among GW veterans in relation to PON1 status, defined by functional PON1_192_ genotype and activity level [29], in a multisite GWI case–control sample. We focused on PON1 status because what determines whether PON1 will protect against a given OP exposure is the catalytic efficiency of PON1 in detoxicating that specific OP compound and the PON1 activity level, which can vary at least 15-fold within a given genotype (Q/Q, Q/R, or R/R) [29]. We also assessed the associations of GWI with a range of GW exposures in PON1 subgroups and explored the possible effects of GW exposure combinations in relation to PON1 status.

## 2. Materials and Methods

### 2.1. Study Design and Participants

This study employed a multisite case–control design to evaluate the association of GW deployment exposures with GWI case status overall and by PON1 status in 398 GW veterans. Coded questionnaire data and previously collected blood samples from three pre-existing GW veteran cohorts were analyzed. The Boston University-based Gulf War Illness Consortium [30] (GWIC cohort, *n* = 258) included GW veterans evaluated at Boston, Houston, and Miami GWIC study sites. The remaining two cohorts were drawn from studies that were based at the San Francisco Veterans Affairs Health Care System (SFVAHCS) but recruited GW veterans nationally. One study was funded by the Department of Veterans Affairs (VA) [31] (SF-VA cohort, *n* = 100 veterans); the other study was funded by the Department of Defense (DOD) [32] (SF-DOD cohort, *n* = 40 veterans). All veterans were deployed to the GW between August 1990 and July 1991. All veterans signed informed consent forms approved by local institutional review boards and the U.S. Army Medical Research and Development Command’s Office of Human Research Protections.

### 2.2. Case Definition

Primary GWI case–control status was determined using the Kansas GWI [33] criteria, as recommended by the IOM/NAM [34]. Briefly, GWI cases were required to have multiple and/or moderate to severe symptoms that had persisted or recurred over six months in at least three of six defined symptom domains: (1) fatigue/sleep problems, (2) pain symptoms, (3) neurological/cognitive/mood symptoms, (4) gastrointestinal symptoms, (5) respiratory symptoms, and (6) dermatological symptoms. Veterans with diagnosed conditions that could account for their chronic symptoms or interfere with their ability to accurately report them (e.g., severe psychiatric disorders) were excluded as GWI cases. Specific exclusionary diagnoses for the current study were identified according to the guidelines established for the GWIC study, as previously described [30]. Veterans with unexplained symptom-defined conditions (e.g., fibromyalgia, chronic fatigue syndrome/myalgic encephalomyelitis, irritable bowel syndrome) were not excluded as Kansas GWI cases. Veterans with insufficient symptoms to meet the GWI case criteria and who reported no exclusionary medical or psychiatric diagnoses were classified as controls.

The veterans from the GWIC and SF-VA cohorts completed the Kansas GW Military and Health questionnaire at the time of research participation. The veterans from the SF-DOD cohort completed the Kansas GW questionnaire 9–15 years after their study participation because the questionnaire was not part of the original SF-DOD protocol.

### 2.3. Evaluation of GW Exposures

Deployment-related exposures were identified using the veterans’ responses to the Kansas GW questionnaire. Veterans were asked to estimate the duration of each exposure reported (i.e., 1–6 days, 7–30 days, 31 days or longer). Veterans were also asked to report whether they had smoked regularly during deployment.

The initial multivariable assessments of the association of GWI with GW exposures indicated that significant risk factors were limited to exposures that potentially affected acetylcholine levels or transmission. Therefore, more detailed assessments of PON1 status in relation to GW exposures focused on exposures with a known or potential effect on acetylcholine or acetylcholinesterase. For the purposes of the study, we considered all variables relevant to wartime exposure to pesticides, PB use, hearing chemical alarms (representing possible nerve agent exposures), and smoking during deployment as GW “cholinergic” exposures. Although some of these exposures (e.g., *N*,*N*-diethyl-meta-toluamide [DEET], PB, and some pesticides) are not known PON1 substrates, we included them based on their potential to act as AChEis and potentially affect acetylcholine levels or PON1 activity.

### 2.4. Blood Collection

Blood samples were obtained from veterans by licensed phlebotomists at the time of their study participation: between 2015 and 2020 (GWIC cohort), 2015 and 2018 (SF-VA cohort), and 2002 and 2006 (SF-DOD cohort). Plasma from the SF-VA and SF-DOD cohorts were collected in BD (Franklin Lakes, NJ, USA) Vacutainer lithium heparin tubes. Plasma from the GWIC cohort were collected in BD Vacutainer sodium heparin tubes. The samples were aliquoted, frozen at −80 °C, and shipped on dry ice to the University of Washington in Seattle for the PON1 assays. The samples from the SF-DOD and SF-VA cohorts were frozen at −80 °C for 1–2 years prior to the PON1 assays. The samples from the GWIC cohort were frozen at −80 °C for up to 3 years prior to the PON1 assays.

### 2.5. PON1 Status Assays

The PON1 status assays were performed on plasma as previously described [35] between 2004 and 2006 (SF-DOD cohort) and 2017 and 2020 (SF-VA and GWIC cohorts). PON1 enzyme activity was evaluated in three substrates: paraoxon (to determine paraoxonase activity), phenyl acetate (for arylesterase activity), and diazoxon (for diazoxonase activity). All assays were run in triplicate using an automated plate reader (Molecular Devices). Samples with replicate values that differed by more than 10% were re-assayed. The initial linear rates of hydrolysis were used for calculations. The results were normalized using the path-length correction software provided by the manufacturer (SoftMax^®^ Pro version 5.4). To retain consistency with earlier studies, the hydrolysis of paraoxon was monitored at 405 nm at 37 °C. Activity was expressed in units/L (U/L) using a molar extinction coefficient of 18 mM^−1^ cm^−1^ for the paraoxon hydrolysis product, p-nitrophenol. The rates of diazoxon hydrolysis were monitored at 270 nm at 25 °C in UV-transparent microplates. Activity was expressed as U/L based on a molar extinction coefficient of 3 mM^−1^ cm^−1^ for the diazoxon hydrolysis product, 2-isopropyl-4-methyl-6-hydroxypyrimidine (IMHP). Arylesterase activity (hydrolysis of phenyl acetate) was monitored at 270 nm at 25 °C in UV-transparent microplates. Activity was expressed in U/mL, based on the molar extinction coefficient of 1.31 mM^−1^ cm^−1^ for phenol. Arylesterase activity was not measured in the SF-DOD sample. To determine PON1 status, data from the participants were separated into three “functional genotype” PON1 activity groups (i.e., QQ, QR, and RR) by plotting the rates of diazoxon hydrolysis (diazoxonase [DZOase]) versus paraoxon hydrolysis (POase) (i.e., DZOase/POase ratio) [29] (see Figure 1). The separation of the three PON1 activity groups was enhanced by carrying the assays out at a high salt concentration. Because the PON1_R192_ alloform is more sensitive to inhibition by high salt levels than the PON1_Q192_ alloform, the use of high salt levels allows for a clearer separation of the functional genotypes.

Twenty-four GW veterans from the SF cohorts had PON1 assayed twice 11–15 years apart because these veterans took part in both the DOD- and the VA-funded studies. We assessed these 24 veterans’ PON1 status and activity levels from both studiesfrom the The other analytic assessments conducted for the current study utilized health data and PON1 assay values from the later time point (SF-VA study) only to coincide with when the veterans’ deployment-related exposure data were obtained.

### 2.6. Data Analyses

Initial analyses compared demographic and military characteristics, PON1 status distribution, and PON1 activity levels between GWI cases and controls. Case–control comparisons utilized chi-square tests for categorical variables. For continuous variables that were normally distributed (Kolmogorov–Smirnov test *p* > 0.05), Student’s *t*-tests were used to compare mean values in cases vs. controls. For non-normal continuous variables, case–control comparisons utilized Wilcoxon–Mann–Whitney nonparametric tests. Spearman’s rank coefficients were used to examine correlations among the deployment-related exposures. All analyses excluded missing values and were performed using SAS version 9.4 (SAS Institute, Inc., Cary, NC, USA).

#### 2.6.1. GW Exposures, GWI, and PON1 Status

Bivariate (unadjusted) associations between GWI case status and all deployment-related exposures were determined by calculating prevalence odds ratios (ORs) and 95% confidence intervals (CIs). Multivariable logistic regression models were used to determine the independent associations of individual exposures with GWI, adjusted for possible confounders, using a backward elimination process. Briefly, the initial models included all demographic, military, and exposure variables that appeared to be significantly associated with case status in the bivariate analyses. Individual variables that were least strongly associated with case status and no longer significant when evaluated with other variables in the model were then sequentially eliminated. For the full sample, the final logistic models identified the adjusted prevalence ORs, controlling for two deployment exposures (the use of skin pesticides and PB pills), age, and rank (officers vs. enlisted ranks).

A similar modeling approach was used to identify independent associations of potentially cholinergic exposures with GWI in the three PON1 status subgroups. The final models evaluated GWI in each PON1 status group in relation to the exposures of interest, controlling for hearing chemical alarms, using skin pesticides, taking PB pills, and rank. Because PON1 status is strongly associated with race, we considered race in all the analyses that evaluated PON1 status. Specifically, race (White/Black/other) was included as a covariate in the PON1 QR and RR strata models. Because there were too few (*n* = 9) non-White veterans with PON1 QQ status and full exposure data to provide valid models that included race, the logistic models for the PON1 QQ subgroup analyses included White/Caucasian veterans only.

#### 2.6.2. Exploratory Analyses

Initial analyses provided preliminary indications that PB use and smoking during deployment potentially modified the effects of other cholinergic exposures in relation to GWI. This suggested the potential for biological synergy among these exposures, similar to effects identified in animal models of GWI [36,37,38]. Therefore, we undertook a series of exploratory analyses to determine whether effects of this type occurred in relation to veterans’ PON1 status. Similar to the hypothesis-testing analyses described above, these exploratory analyses utilized a series of stratified logistic regression models to assess the association of GWI with cholinergic exposures by PON1 status, in veteran subgroups who did and did not take PB pills, and in veteran subgroups who did and did not smoke regularly during deployment. Because of the exploratory nature of these analyses and the small cell size in some of the subgroup analyses, prevalence ORs were determined using exact logistic regression models that included fewer covariates (two GW-related exposures and race) and tested for significance using exact *p*-values.

## 3. Results

### 3.1. Study Population

The study sample included 398 GW veterans: 295 GWI cases and 103 controls. The cases and controls did not differ significantly by sex, race, ethnicity, military branch, or current smoking status. However, the GWI cases were younger, had fewer years of formal education, and included a higher proportion of enlisted personnel (vs. officers) compared to the controls (see Table 1).

### 3.2. PON1 Status and Enzyme Activity Level

Figure 1 shows plots of diazoxonase vs. paraoxonase activities for GWI cases and controls used to determine PON1 status (QQ, QR, RR), as previously described [29]. The graph reflects PON1 activity data from the current study. Table 2 summarizes PON1 status distribution and enzyme activity levels in three different substrates (paraoxonase, arylesterase, and diazoxonase) in relation to GWI case status. PON1 status distribution was nearly identical in GWI cases and controls, and there were no case–control differences in PON1 activity in the three substrates (Table 2). Overall, PON1 activity in each of the three substrates differed by PON1 status, as expected [21,29] (Appendix A). Specifically, RR veterans had higher paraoxonase activity than QQ and QR veterans, while QQ veterans had higher diazoxonase activity than QR and RR veterans. There were no significant PON1 activity differences by GWI case status.

Because PON1 status distribution is known to differ by race [40,41,42], we examined PON1 status by race and ethnicity (see Table 3). PON1 status differed significantly by race in our sample. Consistent with previous reports from general population samples [41,42], significantly more White veterans had QQ status (49%) than Black veterans (7%). Conversely, significantly more Black veterans (50%) had RR status than White (9%) or other/multiracial groups (19%, see Table 3). PON1 status did not differ significantly by Hispanic ethnicity.

### 3.3. Stability of PON1 Status

For the GW veterans from the SF cohorts who had their PON1 status assayed twice 11–15 years apart, all 24 veterans’ PON1 status remained constant over time. In contrast, their PON1 activity levels decreased over time, consistent with most previous reports, which have generally (though not uniformly [43]) identified reduced PON1 activity levels with age [44,45,46] (Appendix A).

### 3.4. Association of Exposures with GWI Case Status

There was significant correlation among deployment-related exposures reported by veterans (Appendix A), particularly among the potentially cholinergic exposures of interest (e.g., hearing chemical alarms, taking PB pills, and using skin pesticides, Spearman’s *ρ* = 0.35 to 0.36, *p* < 0.001). Table 4 summarizes the associations between GWI case status and GW-related exposures. In the unadjusted bivariate analyses, 11 of the GW-related exposures evaluated appeared to be significantly associated with GWI case status. However, after adjusting for other significant exposures, age, and rank during the GW, only the use of skin pesticides, flea collars, and PB pills were significantly and independently associated with GWI in the full sample. Because few veterans (and only one control veteran) reported using flea collars during deployment, this exposure was not included in subsequent multivariable analyses.

The initial analyses provided preliminary indications that GWI risk associated with two cholinergic exposures (nerve agents and skin pesticides) may differ depending on whether veterans used PB or smoked during deployment (Appendix A). In the multivariable models, hearing chemical alarms (reflecting possible nerve agent exposure) and using skin pesticides were significantly associated with GWI among veterans who took PB pills but not among veterans who did not take PB (Appendix A). Similarly, GWI risk in relation to hearing chemical alarms and using skin pesticides were substantially greater among veterans who regularly smoked during deployment compared to veterans who did not smoke (Appendix A). GWI risk associated with other cholinergic exposures did not appear to differ with PB use or deployment smoking status.

### 3.5. GWI Status and Cholinergic Exposures by PON1 Status

Table 5 presents the adjusted ORs for the association of cholinergic exposures with GWI in each PON1 status subgroup. Hearing chemical alarms was significantly associated with GWI among RR veterans (OR = 8.60, *p* = 0.014) but not among QQ or QR veterans. In contrast, using skin pesticides was significantly associated with GWI among QQ (OR = 3.30, *p* = 0.010) and QR (OR = 4.22, *p* < 0.001) veterans but not among RR veterans. There were no other PON1-associated differences in GWI risk in relation to other individual exposures. Of note, smoking during deployment was not identified as an independent risk factor for GWI. However, all RR veterans who reported smoking regularly during deployment were GWI cases. None of the 13 RR control veterans were regular smokers during deployment (Fisher’s exact *p* = 0.074).

### 3.6. Exploratory Analyses: Combined Effects of Cholinergic Exposures in Relation to PON1 Status

The two GWI risk factors that differed by PON1 status in the current study (hearing chemical alarms and using skin pesticides) were also the only two risk factors that initial assessments suggested possible interactive effects with PB use and smoking during deployment (see Appendix A). In light of the presumption that using PB pills would mitigate the serious effects of chemical nerve agents, previous relevant reports [6,47], and the initial findings in the current sample, we conducted exploratory evaluations to determine whether PON1-associated differences in GWI risk factors potentially extended to interactive effects with PB use and smoking.

The first series of exploratory assessments evaluated GWI associations with hearing chemical alarms and using skin pesticides in relation to deployment-related PB use in PON1 status subgroups. In exact logistic models, hearing chemical alarms was not significantly associated with GWI in QQ or QR veterans, regardless of PB use (Table 6). In contrast, hearing chemical alarms was strongly associated with GWI among RR veterans who reported taking PB pills (exact OR = 19.02, *p* = 0.009) but was not associated with GWI in the relatively small group of RR veterans who did not take PB pills (exact OR = 0.97, *p* = 1.00). The identification of hearing chemical alarms as a strong and significant risk factor for GWI only among RR veterans who took PB pills suggested that PB may have modified the effects of nerve agent exposure. Testing in the exact models indicated the chemical alarms x PB interaction term approached statistical significance (exact *p* = 0.080) among RR veterans and was not significant in the other PON1 status subgroups.

A parallel evaluation identified GWI associations with skin pesticides in relation to PB use (Table 6). The exact logistic models determined that among QQ veterans, using skin pesticides was significantly associated with GWI among the veterans who took PB pills (exact OR = 6.34, *p* = 0.001) but not among veterans who did not take PB pills (exact OR = 0.59, *p* = 0.782). For QR veterans, using skin pesticides was significantly associated with GWI regardless of PB use. For RR veterans, using skin pesticides was not significantly associated with GWI regardless of PB use. For QQ veterans, the skin pesticides x PB interaction term in the exact model was highly significant (exact *p* = 0.007).

The second series of exploratory analyses evaluated the same GWI risk factors (hearing chemical alarms and using skin pesticides) in relation to whether veterans reported being regular smokers during deployment (see Table 7). For QQ and QR veterans, GWI risk associated with using skin pesticides was substantially greater among veterans who regularly smoked during deployment (QQ smokers: exact OR = 11.72, *p* = 0.015; QR smokers: exact OR = 20.03, *p* = 0.009) compared to nonsmokers (QQ nonsmokers: exact OR = 2.97, *p* = 0.025; QR nonsmokers: exact OR = 2.89, *p* = 0.019). In the exact models, the skin pesticides x smoking interaction terms approached significance for both QQ (exact *p* = 0.095) and QR (exact *p* = 0.057) veterans. For RR veterans, using skin pesticides was not significantly associated with GWI risk overall or among nonsmokers. Hearing chemical alarms was significantly associated with GWI risk overall and also among nonsmoking veterans. Because all RR veterans who reported smoking during deployment were GWI cases, GWI risk associated with hearing chemical alarms and using skin pesticides were not determined for this subgroup.

## 4. Discussion

The current study assessed associations of GWI with a range of deployment-related exposures in subgroups of GW veterans with different PON1_192_ status (QQ, QR, and RR). The first main finding of the study is that hearing chemical alarms during the war, our proxy for possible exposure to nerve agents, was significantly associated with GWI risk for RR, but not QQ or QR veterans. It is notable that despite differences in methodological approaches, this finding is consistent with previous reports by Haley et al. [26,27] of significantly elevated rates of GWI/Haley syndromes in relation to nerve agent exposure among veteran carriers of the PON1_192_ R allele, particularly RR homozygotes. One difference between this and previous GWI studies is our examination of PON1 status [29], which takes into account both PON1_192_ functional genotype and enzyme activity level (i.e., PON1 phenotype). Previous studies have demonstrated that PON1 status is important for determining the relationship of PON1_192_ polymorphisms with sensitivity to OPs, susceptibilities to disease, and pharmacokinetic status of drug metabolism [48,49]. Furthermore, because of protein truncation mutations, some individuals genotyped as PON1 QR may express enzymes with PON1 QQ or RR phenotypes because only one allele is producing active PON1 [50]. Another methodological difference is our use of the Kansas GWI [33] criteria to determine GWI case status instead of the Haley syndromes [51]. Third, we included race in our consideration of the effects of PON1 status, which is known to vary by race/ancestry [40,41,42] but has not routinely been considered in PON1 studies of GW veterans [26,27,28,52,53].

Investigators have hypothesized that the association between PON1 RR genotype and GWI/Haley syndromes in relation to nerve agent exposure is due to the fact that the PON1_192_ R alloform hydrolyzes sarin less efficiently than the Q alloform [54]. This understanding comes from studies that demonstrated that the PON1_192_ Q alloform hydrolyzes sarin more rapidly than the PON1_192_ R alloform in vitro [24]. Overall, studies have not clearly demonstrated a protective effect of PON1 in relation to nerve agents in vivo. Most in vivo studies have suggested PON1’s capacity to catalyze the hydrolysis of nerve agents is too slow to afford effective protection (i.e., modest affinity, slow rate of turnover, low catalytic efficiency) [55,56], although this is not uniformly the case [57,58].

Nevertheless, multiple studies of veterans, including this one, have identified a significantly increased GWI risk in connection with hearing chemical alarms and other indicators of nerve agent exposure among GW veterans with PON1 RR status or the RR or QR genotype and in veterans with low PON1_192_ Q alloform activity [26,27,28]. This suggests that GW veteran carriers of the PON1_192_ R allele, particularly RR homozygotes, had greater vulnerability to the effects of nerve agents during the Gulf War. Conversely, nerve agent exposure appears not to have posed a risk for veterans with PON1 QQ status/genotype. Whether this is the result of a direct protective effect provided by the PON1_192_ Q alloform or an indirect effect associated with as-yet-unidentified mechanisms remains to be determined. For example, there are other enzymes besides PON1 (e.g., carboxylesterase and butyrylcholinesterase) that can bind and/or react with OP compounds such as sarin and reduce their toxicity [55,59,60]. It will be important for future studies to consider the role of these stoichiometric scavengers in relation to GWI risk.

Beyond looking at PON1 status/genotype and nerve agent exposure, we expanded our evaluation to assess the effects of other GW-related exposures in relation to PON1 status. The second main finding of this study is that using skin pesticides during deployment was significantly associated with GWI risk for QQ and QR status veterans but not for RR veterans. This pattern is consistent with reports that the PON1_192_ Q alloform is less efficient than the PON1_192_ R alloform in protecting against pesticide oxon forms such as chlorpyrifos oxon [61,62,63,64]. We know from government reports that chlorpyrifos and multiple other pesticides were used extensively during the GW [4].

The U.S. Department of Defense has reported that U.S. service members used at least 64 pesticide products during the GW and identified 15 pesticides of potential concern [8]. These included multiple organophosphates (e.g., chlorpyrifos, diazinon, malathion), carbamates (e.g., methomyl, bendiocarb), pyrethroids (e.g., permethrin, d-phenothrin), and the organochlorine delouser lindane. Pesticides and repellants were often used in multiple combinations and for extended periods during the GW [4,8]. The skin pesticide most frequently used by GW veterans was DEET, including a high-concentrate (75%) form no longer used by the military [4,8].

Our analyses indicate that the veterans who self-reported using skin pesticides during deployment were also more likely to report exposure to other types of pesticides (e.g., wearing pesticide-treated uniforms, using flea collars, and witnessing pesticide spraying or fogging). This is consistent with previous reports that GW-related exposures are highly correlated [1,4], that is, that “personnel who reported a high frequency of use with one pesticide form would have been more likely to use (or report) high frequencies for multiple forms and thus might be exposed to a ‘cocktail’ of pesticides” [4]. Thus, it is possible that use of skin pesticides is a proxy of sorts for exposure to multiple types of pesticides/repellants during the GW. There is also suggestive evidence that individuals with the PON1 QQ/QR genotype are more susceptible to developing chronic symptoms in connection with chronic pesticide exposure compared to RR individuals. For example, one study found that QQ/QR farm workers were nearly three times more likely to have symptoms related to chronic pesticide exposure than RR farm workers [64].

The mechanisms that may explain the apparent PON1-associated risks in relation to individual and combined exposures that are not considered PON1 substrates (e.g., PB, DEET) have not been fully elucidated. Previous research has identified factors that may contribute to the types of PON1–exposure associations identified in the current study. For example, smoking has long been known to reduce PON1 activity [65,66,67]. The current study found that QQ/QR status veterans who reported being regular smokers during deployment were at substantially greater GWI risk in relation to using skin pesticides than QQ/QR veterans who were not smokers. This suggests a possible mechanism whereby veterans who smoked during deployment may have experienced a reduction in PON1 activity that potentially exacerbated the already limited capacity for PON1_192_ Q hydrolysis of some pesticides.

Recent research has also provided insights concerning the capacity of carbamate compounds to reduce PON1 activity [68]. Although PB was not specifically tested, physostigmine, a similar compound, and all other carbamates tested were consistently found to inhibit PON1 activity [68]. Our exploratory findings indicate that the use of PB appears to have substantially increased GWI risk associated with hearing chemical alarms among RR veterans and GWI risk associated with skin pesticide use among QQ veterans. Both observations are consistent with a possible mechanism involving a reduction in PON1 activity with PB use that plausibly increased vulnerability to chemical nerve agents among RR veterans and vulnerability to pesticide use in QQ veterans.

More generally, the increased risk for GWI in relation to exposure combinations and interactions identified by the current study is consistent with previous GWI research. Numerous GWI studies have evaluated combined effects of GW deployment-related exposures, primarily in animal models, and have identified effects of combinations of GW-related neurotoxicant exposures that were not seen with individual exposures [1,2]. Several mechanisms have been proposed to explain the increased toxicity and GWI risk that may result from various combinations of GW exposures. The most consistently proposed mechanisms include (1) competitive inhibition of xenobiotic-metabolizing enzymes in the liver and blood, resulting from an excess load or deleterious combination of multiple concurrent toxicants [69,70,71], and (2) changes to blood–brain barrier integrity induced by specific exposures, in conjunction with co-exposures and persistent stress, that allowed increased entry of toxicants into the brain and produced diverse brain alterations, including indicators of a persistent neuroinflammatory state [69,72,73,74].

Relevant to the current findings, numerous animal studies have demonstrated acute and delayed synergistic effects of co-exposure to PB and DEET [36,75,76,77,78]. These include findings in rodent models that combined exposure to PB and DEET produce regional changes in brain AChE activity [79], behavioral deficits [80], and, when combined with stress and/or other pesticides, disruptions in the blood–brain barrier, increased neuroinflammatory markers, and additional brain and behavioral changes [38,81,82,83,84]. Animal studies have also reported that PB provides only slight or no protection from the central cholinergic effects of sarin exposure and can exacerbate sensorimotor deficits [84,85] and increase the production of free radical species/oxidative stress [86] in connection with sarin exposure. Nicotine has also been shown to increase the effects of OP pesticides in animal models, including increased regional expression of nicotinic acetylcholine receptors and AChE activity in the brain and the exacerbation of behavioral deficits [37,87].

Previous studies of GW veterans have also reported greater GWI risk in connection with PB co-exposure to pesticides and chemical nerve agents during the Gulf War. For example, an early study by Haley and colleagues [6] identified “synergistic interaction” between veteran-reported chemical weapons exposure and advanced adverse effects from using PB in relation to significantly increased risk for Haley Syndrome 2 (“confusion-ataxia”). We previously identified an elevated risk for CDC-defined chronic multisymptom illness [39] (CMI) and diminished neuropsychological function among GW veterans who reported high-level exposures to both PB and pesticides during deployment compared to veterans who reported high exposure to either PB or pesticides alone [47]. In a more recent study, we identified significant associations between GWI and combined exposure to both pesticides and PB pills during the GW, as well as the combination of hearing chemical alarms and taking PB pills [88].

In the current study, exploratory analyses that evaluated combined effects of cholinergic exposures in relation to PON1 status provided preliminary evidence that PB may have exacerbated adverse effects of exposure to nerve agents in RR veterans and exposure to pesticides in QQ veterans. Although PB was ordered for use during the Gulf War as a protective measure against potential deadly effects of nerve agent exposure, our findings support earlier indications that PB use may have had unintended consequences. Observed effects were substantially more pronounced in PON1 subgroups than when GWI cases and controls were compared overall. Although preliminary, these findings raise important hypotheses that require further testing in larger studies of GW veterans that are adequately powered to provide a detailed assessment of interactive effects of Gulf War cholinergic exposures in PON1 status subgroups.

As with all studies, the current study has strengths and limitations. Its strengths include the multisite cohort, which included GW veterans from across the country, our utilization of the IOM/NAM panel recommended [34] Kansas GWI [33] case definition, and our “clean” study sample of GWI cases and controls who were unaffected by medical comorbidities that could potentially explain their GWI symptoms. We also evaluated effects of a broader range of GW-related exposures in relation to PON1 status than previous studies and carefully considered the effects of concurrent exposures and race in identifying independent associations with GWI. This study also provided a potentially important innovation by conducting systematic, albeit exploratory, assessments of potential interactive effects of GW cholinergic exposures in relation to PON1 status.

The study’s limitations include our assessment of PON1 activity and status 13–29 years after the GW ended. However, the study’s main findings focus on PON1 status, which, unlike PON1 activity, remains stable over time [44,45,46]. Despite pooling samples from three different cohorts, another limitation is the study’s relatively small sample size. The sample also had a relatively low proportion of minority GW veterans, which limited our ability to conduct detailed assessment of risks associated with PON1 RR status, and to perform multivariable assessment of interactions among cholinergic exposures in this PON1 subgroup. Despite this, it is notable that a number of significant gene–exposure effects were identified in our exploratory assessments that raised potentially important hypotheses for evaluation in future larger studies. Another potential limitation was our reliance on self-reported GW exposures for identifying risk factors for GWI. This issue is common to most GW veteran studies since few exposures of concern were documented during the Gulf War. The validity of self-reported exposures can be diminished by different types and degrees of information bias, leading to over- or under-estimates of risk associations. However, key aspects of our findings lend support to their validity, even when the possible limitations of self-reported exposures are considered. The differences in PON1-associated GWI risk factors identified here were not random but occurred in consistent patterns relative to objective laboratory measures of PON1 status. This is noteworthy because veterans would not have known their PON1 status or differentially reported GW exposures in relation to PON1 status. Moreover, identified differences in PON1 status-exposure associations occurred in the “expected” directions, based on differences in PON1_192_ Q and PON1_192_ R capacity for hydrolyzing nerve agents and pesticides.

## 5. Conclusions

This study both confirmed and extended previous reports indicating that PON1 status/genotype is an important factor in determining associations of GW exposures with GWI. This included corroboration of previous reports that exposures to chemical nerve agents during the GW likely contributed to development of GWI among veterans with PON1 RR status/genotype [26,27]. This study also generated new insights and raised new hypotheses concerning effects of individual and combined GW exposures in relation to PON1 status that require confirmation in future research. These include: (1) findings that use of skin pesticides during the GW is significantly associated with GWI among veterans with QQ and QR PON1 status, but not RR status, and (2) exploratory results that identified potential modulatory effects of both PB and smoking on GWI risk in relation to specific deployment-related cholinergic exposures, which differed by PON1 status. Overall, study results confirm previous indications that the etiology of GWI is complex and involves multiple deployment-related exposures that may differ between veteran subgroups [89].

## Figures and Tables

**Figure 1 ijerph-21-00964-f001:**
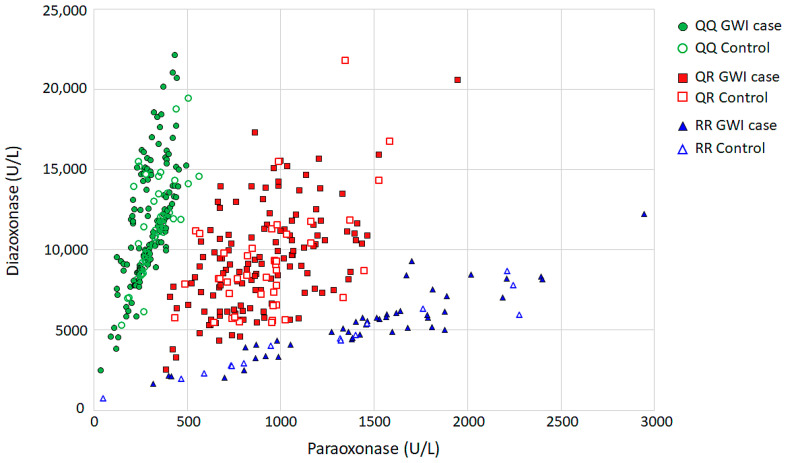
Plot of PON1 diazoxonase vs. paraoxonase activities for GWI cases (solid symbols) and controls (open symbols). The samples are clustered into three discrete groups based on enzyme activity, which correspond closely to QQ (green circles), QR (red squares), and RR (blue triangles) variants of the PON1 Q192R polymorphism.

**Table 1 ijerph-21-00964-t001:** Demographic, military, and health characteristics of study sample.

	All Veterans (*n* = 398)	GWI Cases (*n* = 295)	GW Veteran Controls (*n* = 103)	GWI Cases vs. Controls
Age at time of deployment, mean, (SD), median	26.7 (7.9) 25.2	25.9 (7.1) 24.3	29.1 (9.6) 28.9	*p* < 0.001 *^a^*
Age at time of study, mean, (SD), median	52.4 (6.9) 51.0	51.8 (6.4) 51.0	54.1 (7.9) 54.0	*p* < 0.001 *^a^*
Sex, *n* (%)				
Female	68 (17%)	53 (18%)	15 (15%)	*p* = 0.430 *^b^*
Male	330 (83%)	242 (82%)	88 (85%)	
Race, *n* (%)				
Black/African American	42 (11%)	35 (12%)	7 (7%)	*p* = 0.195 *^b^*
White/Caucasian	324 (82%)	236 (81%)	88 (85%)	
Asian/Pacific Islander	5 (1%)	2 (1%)	3 (3%)	
Native American	2 (<1%)	2 (1%)	0	
Other or multiple races	23 (6%)	18 (6%)	5 (5%)	
Ethnicity, *n* (%)				
Hispanic	35 (9%)	27 (9%)	8 (8%)	*p* = 0.669 *^b^*
Non-Hispanic	363 (91%)	268 (91%)	95 (92%)	
Education level, years, mean, (SD), median	14.8 (2.3) 14.0	14.6 (2.2) 14.0	15.5 (2.5) 16.0	*p* = 0.003 *^a^*
Branch of service in 1990–1991, *n* (%)				
Army	238 (61%)	181 (63%)	57 (55%)	*p* = 0.420 *^c^*
Navy	53 (14%)	36 (12%)	17 (16%)	
Air Force	38 (10%)	25 (9%)	13 (13%)	
Marines	62 (16%)	46 (16%)	16 (16%)	
Military rank in 1990–1991, *n* (%)				
Officer	64 (16%)	30 (10%)	34 (33%)	*p* < 0.001 *^b^*
Enlisted	326 (84%)	257 (90%)	69 (67%)	
Met CDC CMI criteria, *^c^ n* (%)	333 (84%)	287 (97%)	46 (45%)	*p* < 0.001 *^b^*
Regular smoker at time of study, *n* (%)	35 (9%)	28 (10%)	7 (7%)	*p* = 0.376 *^b^*

Abbreviations: GWI = Gulf War illness, SD = standard deviation. *^a^* Wilcoxon–Mann–Whitney test. *^b^* chi-square test. *^c^* Centers for Disease Control and Prevention (CDC) criteria for Chronic Multisymptom Illness (CMI) [39].

**Table 2 ijerph-21-00964-t002:** PON1 status and enzyme activity by GWI case–control status.

	All Veterans (*n* = 398)	GWI Cases (*n* = 295)	GW Veteran Controls (*n* = 103)	GWI Cases vs. Controls
**PON1 status distribution**	*n* (%)	*n* (%)	*n* (%)	
QQ	169 (42%)	125 (42%)	44 (43%)	*p* = 0.840 *^b^*
QR	172 (43%)	126 (43%)	46 (45%)	
RR	57 (14%)	44 (15%)	13 (13%)	
**PON1 enzyme activity *^a^* (units/mL) by substrate** [mean, (median)]				
Paraoxonase	723.2 (646.7)	725.2 (642.6)	717.3 (667.9)	*p* = 0.797 *^d^*
Arylesterase	134.8 (132.3)	134.0 (129.7)	137.2 (132.5)	*p* = 0.569 *^c^*
Diazoxonase	9658.8 (9315.2)	9690.0 (9407.5)	9569.5 (9263.7)	*p* = 0.783 *^c^*

Abbreviations: GWI = Gulf War illness. *^a^* in designated substrate. *^b^* chi-square test. *^c^ t*-test. *^d^* Wilcoxon–Mann–Whitney test.

**Table 3 ijerph-21-00964-t003:** PON1 status distribution by race and ethnicity.

	*n*	PON1 Status			Race/Ethnicity Comparison
		QQ *n* (%)	QR *n* (%)	RR *n* (%)	
**Race** (All Categories)					*p* < 0.001
Black/African American	42	3 (7%)	18 (43%)	21 (50%)	
White/Caucasian	324	160 (49%)	134 (41%)	30 (9%)	
Asian/Pacific Islander	5	0	4 (80%)	1 (20%)	
Native American	2	0	2 (100%)	0	
Other/Multiracial	23	6 (26%)	12 (52%)	5 (22%)	
**Race Group** (Three Strata)					*p* < 0.001
White/Caucasian	324	160 (49%)	134 (41%)	30 (9%)	
Black/African American	42	3 (7%)	18 (43%)	21 (50%)	
All Other/Multiracial	31	6 (19%)	19 (61%)	6 (19%)	
**Ethnicity**					*p* = 0.548
Hispanic	35	12 (34%)	18 (51%)	5 (14%)	
Non-Hispanic	363	157 (43%)	154 (42%)	52 (14%)	

**Table 4 ijerph-21-00964-t004:** Association of deployment exposures with GWI case status.

Deployment Experience/Exposure	GWI Cases (*n* = 295)		Controls (*n* = 103)		OR (95% CI) (Unadjusted)	OR *^a^*(95% CI) (Adjusted)
Saw oil fire smoke						
Ever	250	(87%)	79	(77%)	1.97 (1.10, 3.51) *^b^*	1.12 (0.57, 2.21)
>7 days	186	(65%)	57	(56%)	1.44 (0.92, 2.30)	1.00 (0.59, 1.67)
Heard chemical alarms sound						
Ever	247	(86%)	65	(64%)	3.51 (2.08, 5.94) *^c^*	1.66 (0.88, 3.16)
>7 days	136	(47%)	27	(27%)	2.50 (1.52, 4.11) *^c^*	1.71 (0.98, 2.98)
Within 1 mile of exploding SCUD missile						
Ever	143	(50%)	37	(36%)	1.76 (1.10, 2.80) *^b^*	1.44 (0.85, 2.44)
>7 days	45	(16%)	8	(8%)	2.19 (1.00, 4.83)	2.39 (0.97, 5.86)
Directly involved in ground combat						
Ever	130	(45%)	30	(29%)	2.03 (1.25, 3.29) *^b^*	1.08 (0.60, 1.94)
>7 days	54	(19%)	13	(13%)	1.61 (0.84, 3.09)	0.92 (0.44, 1.95)
Contact with POW						
Ever	165	(58%)	40	(39%)	2.15 (1.36, 3.40) *^b^*	1.15 (0.65, 2.03)
>7 days	92	(32%)	20	(19%)	1.97 (1.10, 3.40) *^b^*	1.14 (0.61, 2.13)
Contact with destroyed enemy vehicles						
Ever	200	(70%)	51	(50%)	2.30 (1.45, 3.65) *^c^*	1.09 (0.61, 1.94)
>7 days	115	(40%)	23	(23%)	2.30 (1.36, 3.87) *^b^*	1.07 (0.58, 1.98)
Used pesticide cream/spray on skin						
Ever	210	(73%)	37	(36%)	4.86 (3.01, 7.86) *^c^*	3.90 (2.30, 6.60) *^c^*
>7 days	188	(66%)	28	(27%)	5.09 (3.09, 8.37) *^c^*	4.24 (2.46, 7.31) *^c^*
Wore uniform treated with pesticides						
Ever	138	(48%)	26	(25%)	2.76 (1.67, 4.56) *^c^*	1.54 (0.84, 2.84)
>7 days	120	(42%)	20	(19%)	3.00 (1.75, 5.16) *^c^*	1.64 (0.86, 3.13)
Wore flea collars						
Ever	37	(13%)	1	(1%)	15.16 (2.05, 111.95) *^c^*	9.82 (1.26, 76.56) *^b^*
Saw area sprayed/fogged with pesticides						
Ever	106	(38%)	22	(21%)	2.24 (1.32, 3.81) *^b^*	1.28 (0.69, 2.35)
>7 days	62	(22%)	11	(11%)	2.38 (1.20, 4.72) *^b^*	1.42 (0.66, 3.06)
Took PB (NAPP) pills						
Ever	239	(83%)	62	(60%)	3.29 (1.99, 5.49) *^b^*	1.97 (1.11, 3.49) *^b^*
>7 days	154	(54%)	29	(28%)	2.95 (1.81, 4.81) *^b^*	2.03 (1.20, 3.46) *^b^*
Regular smoker during deployment	66	(23%)	20	(19%)	1.25 (0.71, 2.19)	0.91 (0.49, 1.72)

Abbreviations: GWI = Gulf War illness; OR = odds ratio; CI = confidence interval; POW = prisoners of war; PB = pyridostigmine bromide; NAPP = nerve agent pyridostigmine pretreatment. *^a^* Logistic regression, adjusted for using pesticide cream/spray on skin, taking PB pills, age, and rank. Significant associations: *^b^ p* < 0.05; *^c^ p* < 0.001.

**Table 5 ijerph-21-00964-t005:** Association of GWI case status with potentially cholinergic exposures by PON1 status.

	QQ PON1 Status *^a^* (*n* = 156:113 GWI Cases, 43 Ctrl)			QR PON1 Status (*n* = 167:121 GWI Cases, 46 Ctrl)			RR PON1 Status (*n* = 56:43 GWI Cases, 13 Ctrl)		
	*n* (%) Exposed			*n* (%) Exposed			*n* (%) Exposed		
** Exposures **	GWI	Ctrl	OR_adj_ *^a^* (95% CI)	GWI	Ctrl	OR_adj_ *^b^* (95% CI)	GWI	Ctrl	OR_adj_ *^b^* (95% CI)
Heard chemical alarms sound	95 (84%)	26 (60%)	1.87 (0.68, 5.09)	107 (88%)	34 (74%)	0.92 (0.30, 2.83)	37 (86%)	5 (38%)	8.60 (1.54, 47.92) *^d^*
Used pesticide cream/spray on skin	83 (73%)	15 (34%)	3.30 (1.33, 8.22) *^d^*	89 (73%)	17 (37%)	4.22 (1.89, 9.44) *^e^*	31 (72%)	5 (38%)	1.50 (0.24, 9.50)
Saw area sprayed/fogged w/pesticides	38 (35%)	9 (20%)	0.91 (0.32, 2.60)	50 (42%)	10 (22%)	1.81 (0.75, 4.36)	12 (29%)	3 (23%)	1.02 (0.15, 7.16)
Wore uniform treated with pesticides	55 (49%)	12 (27%)	1.22 (0.45, 3.32)	59 (48%)	11 (24%)	1.92 (0.76, 4.89)	18 (42%)	3 (23%)	1.30 (0.24, 7.02)
Took PB (NAPP) pills	95 (84%)	23 (52%)	2.60 (0.98, 6.86) *^c^*	104 (85%)	30 (65%)	2.02 (0.74, 5.53)	35 (81%)	9 (69%)	1.01 (0.16, 6.45)
Regular smoker during deployment	27 (24%)	9 (20%)	0.76 (0.28, 24.33)	29 (24%)	11 (24%)	0.75 (0.30, 1.86)	9 (21%)	0	undefined

Abbreviations: GWI = Gulf War illness, Ctrl = controls; OR = odds ratio, CI = confidence interval, PB (NAPP) = pyridostigmine bromide (nerve agent pyridostigmine pretreatment). *^a^* PON1 QQs: evaluated in White veterans only; logistic regression models adjusted for hearing chemical alarms, using pesticide cream/spray on skin, taking PB pills, and rank. *^b^* Logistic regression models adjusted for hearing chemical alarms, using pesticide cream/spray on skin, taking PB pills, rank, race (White/Black/other). *^c^* Trend for association (0.05 < *p* < 0.10). Significant associations: *^d^ p* < 0.05; *^e^ p* < 0.001.

**Table 6 ijerph-21-00964-t006:** Exploratory assessment of association of GWI with hearing chemical alarms and use of skin pesticides in PON1 status subgroups: Effects of taking PB pills.

QQ PON1 Status	All QQ*^a^* (*n* = 156: 113 GWI Cases, 43 Ctrl)			QQ: Did Not Take PB Pills (*n* = 39: 18 GWI Cases, 21 Ctrl)			QQ: Took PB Pills (*n* = 117: 95 GWI Cases, 22 Ctrl)		
Exposures	*n* (%) Exposed		ORadj *^a^* (95% CI)	*n* (%) Exposed		Exact ORadj *^a,b^*	*n* (%) Exposed		Exact ORadj *^a,b^*
	GWI	Ctrl		GWI	Ctrl		GWI	Ctrl	
Heard chemical alarms sound	95 (84%)	26 (60%)	2.00 (0.84, 4.79)	9 (50%)	9 (43%)	OR = 1.59, *p* = 0.745	86 (91%)	17 (77%)	OR = 2.06, *p* = 0.456
Used pesticide on skin	83 (73%)	15 (34%)	4.17 (1.87, 9.28) *^e^*	4 (22%)	6 (29%)	OR = 0.59, *p* = 0.782	79 (83%)	9 (39%)	OR = 6.34, *p* = 0.001 *^d^*
**QR PON1 status**	**All QRs** **(*n* = 167: 121 GWI Cases, 46 Ctrl)**			**QR: Did not take PB pills** **(*n* = 34: 18 GWI Cases, 16 Ctrl)**			**QR: Took PB pills** **(*n* = 133: 103 GWI Cases, 30 Controls)**		
Exposures	*n* (%) Exposed		ORadj *^c^*(95% CI)	*n* (%) Exposed		Exact ORadj *^b,c^*	*n* (%) Exposed		Exact ORadj *^b,c^*
	GWI	Ctrl		GWI	Ctrl		GWI	Ctrl	
Heard chemical alarms sound	107 (88%)	34 (74%)	1.44 (0.56, 3.68)	9 (50%)	7 (44%)	OR = 0.36, *p* = 0.476	98 (94%)	27 (90%)	OR = 1.39, *p* = 0.942
Used pesticide on skin	89 (73%)	17 (37%)	4.21 (1.97, 8.98) *^e^*	10 (56%)	2 (12%)	OR = 10.70, *p* = 0.022 *^d^*	79 (76%)	15 (50%)	OR = 3.05, *p* = 0.018 *^d^*
**RR PON1 status**	**All RRs** **(*n* = 56: 43 GWI Cases, 13 Ctrl)**			**RR: Did not take PB pills** **(*n* = 12: 8 GWI Cases, 4 Ctrl)**			**RR: Took PB pills** **(*n* = 44: 35 GWI Cases, 9 Ctrl)**		
Exposures	*n* (%) Exposed		ORadj *^c^* (95% CI)	*n* (%) Exposed		Exact ORadj *^b,c^*	*n* (%) Exposed		Exact ORadj *^b,c^*
	GWI	Ctrl		GWI	Ctrl		GWI	Ctrl	
Heard chemical alarms sound	37 (86%)	5 (38%)	7.18 (1.45, 35.52) *^d^*	5 (62%)	2 (50%)	OR = 0.97, *p* = 1.00	32 (91%)	3 (33%)	OR = 19.02, *p* = 0.009 *^d^*
Used pesticide on skin	31 (72%)	5 (38%)	2.03 (0.43, 9.57)	3 (37%)	1 (25%)	OR = 2.12, *p* = 1.00	28 (80%)	4 (44%)	OR = 2.19, *p* = 0.695

Abbreviations: GWI = Gulf War illness; Ctrl = controls; PB = pyridostigmine bromide; OR = odds ratio; CI = confidence interval. *^a^* QQ PON1 subgroup evaluated in White veterans only; logistic regression models adjusted for hearing chemical alarms, using pesticide cream/spray on skin. *^b^* Exact logistic regression models and *p* values. *^c^* QR and RR PON1 subgroups: Logistic regression models adjusted for hearing chemical alarms, using pesticide cream/spray on skin, race (White vs. all other). Significant associations: *^d^ p* < 0.05; *^e^ p* < 0.001.

**Table 7 ijerph-21-00964-t007:** Exploratory assessment of association of GWI with hearing chemical alarms and use of skin pesticides in PON1 status subgroups: Effects of smoking during deployment.

QQ PON1 Status	All QQs *^a^* (*n* = 155:111 GWI Cases, 44 Ctrl)			QQ: Not Regular Smoker (*n* = 119:84 GWI Cases, 35 Ctrl)			QQ: Regular Smoker (*n* = 36:27 GWI Cases, 9 Ctrl)		
Exposures	*n* (%) Exposed		ORadj *^a^* (95% CI)	*n* (%) Exposed		Exact ORadj *^a,b^*	*n* (%) Exposed		Exact ORadj *^a,b^*
	GWI	Ctrl		GWI	Ctrl		GWI	Ctrl	
Heard chemical alarms sound	95 (84%)	26 (60%)	2.00 (0.84, 4.79)	68 (81%)	21 (62%)	OR = 1.68, *p* = 0.404	25 (93%)	5 (56%)	OR = 5.11, *p* = 0.348
Used pesticide on skin	83 (73%)	15 (34%)	4.17 (1.87, 9.28) *^e^*	58 (69%)	13 (37%)	OR = 2.97, *p* = 0.025 *^d^*	23 (85%)	2 (22%)	OR = 11.72, *p* = 0.015 *^d^*
**QR PON1 status**	**All QRs** **(*n* = 167:121 GWI Cases, 46 Ctrl)**			**QR: Not Regular Smoker** **(*n* = 128:93 GWI Cases, 35 Ctrl)**			**QR: Regular Smoker** **(*n* = 40:29 GWI Cases, 11 Ctrl)**		
Exposures	*n* (%) Exposed		ORadj *^c^*(95% CI)	*n* (%) Exposed		Exact ORadj *^b,c^*	*n* (%) Exposed		Exact ORadj *^b,c^*
	GWI	Ctrl		GWI	Ctrl		GWI	Ctrl	
Heard chemical alarms sound	107 (88%)	34 (74%)	1.44 (0.56, 3.68)	79 (85%)	27 (77%)	OR = 1.12, *p* = 1.00	28 (97%)	7 (64%)	OR = 3.52, *p* = 0.644
Used pesticide on skin	89 (73%)	17 (37%)	4.21 (1.97, 8.98)*^e^*	65 (70%)	15 (43%)	OR = 2.89, *p* = 0.019 *^d^*	24 (83%)	2 (18%)	OR = 20.03, *p* = 0.009 *^d^*
**RR PON1 status**	**All RRs** **(*n* = 56:43 GWI Cases, 13 Ctrl)**			**RR: Not Regular Smoker** **(*n* = 47:34 GWI Cases, 13 Ctrl)**			**RR: Regular Smoker** **(*n* = 9:9 GWI Cases, 0 Ctrl)**		
Exposures	*n* (%) Exposed		ORadj *^c^* (95% CI)	*n* (%) Exposed		Exact ORadj *^b,c^*	*n* (%) Exposed		Exact ORadj
	GWI	Ctrl		GWI	Ctrl		GWI	Ctrl	
Heard chemical alarms sound	37 (86%)	5 (38%)	7.18 (1.45, 35.52)*^d^*	29 (85%)	5 (38%)	OR = 6.35, *p* = 0.044 *^d^*	8 (89%)	0	-undefined-
Used pesticide on skin	31 (72%)	5 (38%)	2.03 (0.43, 9.57)	23 (68%)	5 (38%)	OR = 1.43, *p* = 0.944	8 (89%)	0	-undefined-

Abbreviations: GWI = Gulf War illness; Ctrl = controls; PB = pyridostigmine bromide; OR = odds ratio; CI = confidence interval. *^a^* PON1 QQ subgroup evaluated in White veterans only; logistic regression models adjusted for hearing chemical alarms, using pesticide cream/spray on skin. *^b^* Exact logistic regression models and *p* values. *^c^* QR and RR PON1 subgroups: Logistic regression models adjusted for hearing chemical alarms, using pesticide cream/spray on skin, race (White vs. all other). Significant associations: *^d^ p* < 0.05; *^e^ p* < 0.001.

## Data Availability

The datasets generated during/or analyzed during the current study are available from the corresponding author on reasonable request.

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
