# Peer review of "PON1 Status in Relation to Gulf War Illness: Evidence of Gene–Exposure Interactions from a Multisite Case–Control Study of 1990–1991 Gulf War Veterans"

_ijerph, 2024, doi:10.3390/ijerph21080964_

Round 1

Reviewer 1 Report

Comments and Suggestions for Authors

I think the article is interesting. I have a question: how long have the samples been frozen at -80º?.
Maybe they should put the time they have been frozen.

Author Response

Comment 1: I think the article is interesting. I have a question: how long have the samples been frozen at -80º? Maybe they should put the time they have been frozen.

Response 1: The samples were frozen at −80°C for different amounts of time.  Samples from the SF-DOD and SF-VA cohorts were frozen for 1-2 years prior to PON1 assay.  Samples from the GWIC cohort were frozen from <1 to 3 years prior to PON1 assays.  We have added this information to the methods section at the bottom of page 3 and the top of page 4: “Samples from the SF-DOD and SF-VA cohorts were frozen at −80°C for 1-2 years prior to PON1 assays. Samples from the GWIC cohort were frozen at −80°C for up to 3 years prior to PON1 assays.”

Reviewer 2 Report

Comments and Suggestions for Authors

Overall, the work is well written and presented, providing crucial information that significantly contributes to our understanding of Gulf War Illness. It is suggested to make the following changes so that the work is ready to be published:

The data in Figure 1 is unclear if they are original or if the graph was previously published; please clarify. If it is an original analysis, it should be placed outside the methodology section but in the results section. Clearly state this and cite the original source if it is a previously published graphic.

It would be appreciated if the change in the design or format of the tables could be considered since some are very loaded with information, and the columns cannot be precisely distinguished.

A significant association was found between pesticide exposure and Gulf War Illness. It is suggested that examples of the pesticides to which the experimental subjects were exposed be mentioned in the discussion.

Author Response

Comment 1: The data in Figure 1 is unclear if they are original or if the graph was previously published; please clarify. If it is an original analysis, it should be placed outside the methodology section but in the results section. Clearly state this and cite the original source if it is a previously published graphic.

Response: Data in figure 1 is original and reflects PON1 activity data from the current study.  This has been clarified and specifically stated in the manuscript.  We have also moved figure 1 to the results section. Please note that moving the figure necessitated some changes to the figure legend, which previously included information that related to methods. That text has now been moved to the methods section.

Comment 2: It would be appreciated if the change in the design or format of the tables could be considered since some are very loaded with information, and the columns cannot be precisely distinguished.

Response: We agree with the referee. Some tables appear to have been resized when converted to the journal format. Those tables have now been re-formatted to be clearer.

Comment 3: A significant association was found between pesticide exposure and Gulf War Illness. It is suggested that examples of the pesticides to which the experimental subjects were exposed be mentioned in the discussion.

Response:  We agree it is helpful and informative to provide examples of the types of pesticides to which Gulf War veterans were exposed during deployment. The original manuscript offered some details regarding the types of pesticides to which Gulf War veterans were exposed (e.g., chlorpyrifos, lindane, pyrethroids, flea collars, DEET).

The study questionnaire asked veterans about their exposure to four general categories of pesticides (e.g., pesticide cream/spray on the skin, area fogged with pesticides), but did not ask about specific compounds/chemicals, since previous reports [e.g., refs 4,8] have shown that veterans were usually not aware of the specific chemicals in the pesticides they used.  A 2003 DOD report found that, overall, U.S. servicemembers had used at least 64 pesticide products during the GW, which contained 37 active ingredients, and identified 15 pesticides of potential concern [ref 8].

We have now expanded the section of the discussion related to types of pesticide exposures to provide an overview of this information. We also provide multiple examples of the specific types of pesticides to which GW veterans were exposed on page 13 of the discussion:

“The U.S. Department of Defense has reported that U.S. servicemembers used at least 64 pesticide products during the GW, and identified 15 pesticides of potential concern [8]. These included multiple organophosphates (e.g., chlorpyrifos, diazinon, malathion), carbamates (e.g., methomyl, bendiocarb), pyrethroids (e.g., permethrin, d-phenothrin), and the organochlorine delouser, lindane. Pesticides and repellants were often used in multiple combinations and for extended periods during the GW [4,8]. The skin pesticide most frequently used by GW veterans was DEET, including a high concentrate (75%) form no longer used by the military [4,8].”

Reviewer 3 Report

Comments and Suggestions for Authors

General comment:

This article evaluated associations of GWI of deployment-related exposures in GW veterans with different PON1 status during Gulf War. Overall, it is well-written and easy to follow up. However, there are room for improvement to make it more accessible. I offer several comments and feedback that I hope will be helpful to enhance the quality of the article.

Major comments:

·       In initial analyses, the authors performed multiple analyses based on categorical and continuous variables to see any statistical differences or associations between case and control. I was wondering how the normality assumption was verified. Any chance to put more elaborate this work, such as normality test, generation of quantile-quantile plot? Obviously, the authors need to consider different test if assumptions not met.

·       In Table 4, can the authors double check to see if the adjusted OR in heard chemical alarms sound is really, correct? Typically, the estimated OR is attenuated with adjusting covariates in the model, and these patterns were observed by other deployment experience/exposure variables. Moreover, how the authors had unadjusted OR when the count is sparse, especially in wearing uniform treated with pesticides? It is very high and wide 95% confidence intervals. What test was applied in such case? If different test was used, this should be noted in somewhere.

·       The authors should have reported quantities consistently. In Table 6, p-value was reported in exact OR, rather than having 95% confidence interval. Wouldn’t be better to keep persistently format in all tables? The models should be able to provide the estimate, 95% confidence interval, and p-value.

·       Looking at Table 2, overall veterans in RR were almost 1/3 of the other PON1 status, resulting in underpower of the analysis. Can the authors have any comments on this potential issue?

Minor comments:

·       In Figure 1, each case and control need to present at different colors. The current presentation is hard to see any separation of the three groups.

·       Tables 5 to 7 should present a better way for potential readers.

·       In Section 3.6, the authors consistently introduced new term, exact logistic models and exact OR. Can the authors clarify how these were different from regular logistic models and OR?

Author Response

Major Comment 1: In initial analyses, the authors performed multiple analyses based on categorical and continuous variables to see any statistical differences or associations between case and control. I was wondering how the normality assumption was verified. Any chance to put more elaborate this work, such as normality test, generation of quantile-quantile plot? Obviously, the authors need to consider different test if assumptions not met.

Response:  For our case-control comparisons, evaluation of the normality assumption is applicable to the use of parametric t-tests when comparing means of the continuous variables assessed. We assessed relatively few continuous variables: Only three in the initial descriptive table (age at deployment, age at study, education level) and three PON1 enzyme activity levels reported in Table 2. Some continuous variables were normally distributed, others were not.  There is an informative literature and common view that, per the central limit theorem, the normality assumption is not essential when comparing means in larger samples (variously identified as sample sizes of n >30 to sample sizes of n >100).  Therefore, mean values of continuous variables are commonly compared using t-tests in larger samples, even when the data are not normally distributed.  Mean value comparisons are also familiar and easily interpretable for readers, so we considered mean values and t test comparisons to be the most straightforward method for comparing continuous variables in cases and controls in the manuscript. Further, when we used nonparametric tests to compare distributions of continuous variables in the two populations (i.e., cases and controls), we obtained similar results.  Because of the concern raised by the reviewer, we now specify in the methods section that normality of continuous variables were evaluated using the Kolmogorov-Smirnov test. We also now provide results of parametric tests for case/control comparisons of normally distributed continuous variables and non-parametric tests for non-normal continuous variables, and report both means (with standard deviations) and median values in the tables.

Assumptions regarding normal distributions did not affect any other case/control comparison analyses done for the paper.  For example, chi square testing was used in comparing categorical variables in cases vs. controls. And multivariable logistic regression models, which were used to identify independent associations of exposure variables with our binary case/control outcomes, do not rely on normality assumptions.

Major Comment 2: In Table 4, can the authors double check to see if the adjusted OR in heard chemical alarms sound is really, correct? Typically, the estimated OR is attenuated with adjusting covariates in the model, and these patterns were observed by other deployment experience/exposure variables. Moreover, how the authors had unadjusted OR when the count is sparse, especially in wearing uniform treated with pesticides? It is very high and wide 95% confidence intervals. What test was applied in such case? If different test was used, this should be noted in somewhere.

Response: Thank you for this question, but we are not certain what the reviewer is asking. In Table 4, the point estimate for the adjusted OR for GWI in relation to hearing chemical alarms (ORadj = 1.66, ns) is lower than the unadjusted OR (OR = 3.51, p<0.001). So, as mentioned by the reviewer, the magnitude of association was attenuated when covariates were considered in the adjusted model.  This pattern is similar to other exposures evaluated in Table 4, as also mentioned by the reviewer. So, it appears the adjusted OR results presented in Table 4 are consistent with the reviewer’s expectation. 

The question regarding how we were able to generate unadjusted ORs when the count is sparse seems counterintuitive, since small cell size is more typically a concern with adjusted ORs/multivariable analyses. In any case, there were not sparse counts/small cell sizes in the assessment of ORs for wearing uniforms treated with pesticides. 

As suggested, we double checked the analytic results for the OR estimate for the association of chemical alarms with GWI case status (adjusted for using skin pesticides, taking PB pills, age, and rank). Our re-analysis indicated the adjusted OR is correct (i.e., there should be no change from the ORadj presented in Table 4).

Because the expectations outlined by the reviewer were consistent with the results we presented, we are unsure if we correctly interpreted the reviewer’s points. Or if perhaps the reviewer was misled by problems with reading the tables, as 2 reviewers pointed out. Please let us know if we have adequately addressed the points made by the reviewer.

Major Comment 3: The authors should have reported quantities consistently. In Table 6, p-value was reported in exact OR, rather than having 95% confidence interval. Wouldn’t be better to keep persistently format in all tables? The models should be able to provide the estimate, 95% confidence interval, and p-value.

Response: We understand it can sometimes be confusing when two different modeling methods are used in the same paper or same table. In this case, however, we were addressing two different types of research questions that required 2 logistic regression approaches in order to provide the most valid and informative analytic results. It is unclear if the reviewer was aware that differences in how modeling results are presented in some tables are linked to the different analytic approaches used for hypothesis testing and exploratory analyses.

We had hoped that the text in the Methods section clearly outlined our rationale and approach for both: (1) hypothesis-testing analyses that assessed independent associations of exposures with GWI using standard logistic regression, and (2) exploratory analyses to address additional questions raised by our initial results (sometimes with small cell sizes) using exact logistic regression. Exploratory analyses are commonly used in population health research as a first step to identify previously unknown findings and to raise hypotheses for further testing.  

As described in the Methods section, a standard multivariable logistic regression modelling approach was used to address our primary a priori study questions concerning associations between Gulf War exposures and GWI—overall, and in PON1 status subgroups. The study sample provided adequate power to test hypotheses for each exposure of interest. Modelling results are shown in Tables 4 and 5, and presented as adjusted ORs, with 95% confidence intervals. 

Our initial assessment of terms for these models indicated that two “cholinergic exposures” defined for the study (i.e., PB use and smoking during deployment) may modify effects of other cholinergic exposures in ways that were potentially linked to PON1 status. Because these associations had not previously been identified and were not a priori study questions or hypotheses, we determined that exploratory evaluations were warranted to carefully identify associations that occurred in our data.

This required assessments among veteran subgroups defined by the presence/absence of 2 exposures and by PON1 status, so some subgroups were relatively small. To optimize determination of valid associations in these subgroups, we utilized exact logistic regression for the multivariable models. This method provides point estimates for exact ORs, exact confidence intervals (C.I.s), and exact p values.  As explained in the methods, results for these analyses were reported as exact ORs and exact p values. While both C.I.s and p values provide an indication of statistical significance, exact p values provide a more easily interpreted indicator of the degree of significance. We therefore considered them more informative for the “exploratory” purposes of these analyses.   

Specifically, exact logistic regression was used for exploratory analyses undertaken to evaluate the association of GWI with hearing chemical alarms and using skin pesticides in subgroups defined by a) PB use vs. no PB use and b) smoking during deployment vs. not smoking.  Results are shown in Table 6 (PB use vs. no PB use) and Table 7 (smoking in theater vs. not smoking), and presented as ‘Exact ORs’ and ‘exact p-values’. 

In general, we would concur with the reviewer’s suggestion that analytic results be provided in a consistent way in different tables, when those results reflect the same types of research questions and analyses. However, in this paper, Tables 4 and 5, and the first columns of Tables 6 and 7 present multivariable results that test specific hypotheses using standard logistic regression models.  In contrast, the subgroup analyses in Tables 6 and 7 present results of exploratory analyses obtained using exact logistic regression models.  We believe it provides clarity and more valid results to present the exact ORs and p values for these exploratory analyses and sets them apart from the hypothesis testing results. 

Based on the reviewer’s comment, we wonder if the distinction between the hypothesis-testing analyses and exploratory analyses were not sufficiently clear in the paper.  Therefore, in addition to the existing description of the two types of analyses in the Methods section, we have (1) added clarifying wording regarding use of exploratory assessments and exact testing to the titles and footnotes of Tables 6 and 7, and (2) added clarifying wording regarding use of exploratory assessments and exact testing methods to the abstract.

Major Comment 4: Looking at Table 2, overall veterans in RR were almost 1/3 of the other PON1 status, resulting in underpower of the analysis. Can the authors have any comments on this potential issue?

Response:  It is unclear if the reviewer’s concern relates specifically to statistical power for Table 2 analysis.  We are not aware of any assumption or requirement for subgroups (here, PON1 status subgroups) to be comparably sized when using chi square tests to compare categorical variables between two populations (here, cases and controls). Such analyses often involve differently sized subgroups, including individual subgroups that are a fraction of the size of other subgroups. Cell size is primarily a consideration only if the expected size of a given cell is < 5.  Other considerations might include the overall sample size, but the sample used in Table 2 (n=398) substantially exceeds any recommended minimum sample required.  Considering the analysis presented in Table 2, we see no indication that the sample is underpowered to identify significant differences in the distribution of QQ, QR, and RR PON1 status among GWI cases and controls. 

The relatively small proportion of RR veterans does have implications for other study findings, however. This is described in both the Results and Discussion sections of the paper, particularly as relates to PON1 subgroup analyses associated with smoking (Table 5) and combinations of cholinergic exposures (Tables 6 and 7). It was further described as a study limitation in the Discussion section, specifically in relation to the study’s limited capacity for assessment of effects of exposures combinations in RR status veterans.

Minor Comment 1: In Figure 1, each case and control need to present at different colors. The current presentation is hard to see any separation of the three groups.

Response: Thank you for this suggestion; we now present the different PON1 groups in different colors in Figure 1.

Minor Comment 2: Tables 5 to 7 should present a better way for potential readers.

Response: We agree with the reviewer and have re-formatted tables 5-7 for easier comprehension. The tables appear to have been resized when converted to the journal format.

Minor Comment 3: In Section 3.6, the authors consistently introduced new term, exact logistic models and exact OR. Can the authors clarify how these were different from regular logistic models and OR?

Response: Exact logistic regression is routinely used to provide multivariable assessment of associations of independent variables with binary outcomes in smaller samples.  It is useful and appropriate for small-sample situations that can result in failure (nonconvergence) with maximum likelihood estimation methods used by standard logistic regression. Parameter estimates do not rely on asymptotic assumptions; tests of significance are based on exact methods. Exact logistic regression procedures are commonly available in widely used statistical software packages (e.g., Stata, R, SAS).

It is unclear if the reviewer is suggesting that we provide explanatory information about this method in the paper. In our experience, this method is familiar to epidemiologists and biostatisticians who conduct population health research.